# Screening and targeted sequencing of stool for microbiologic confirmation and drug resistance determination in paucibacillary tuberculosis

Tara E. Ness[1,2☯]*, Mangaliso Ziyane[3,4☯], Nontobeko Maphalala[5], Abigail Seeger[1], Anca Vasiliu[1,6,7], Wethusonke Khumalo[5], Mdigo Thunzini[3], Sindisiwe Dlamini[3], Gugu Maphalala[3], Clement Gascua[5], Christoph Lange[1,6,7,8], Samantha Meyer[4], Bryce Inman[2], Viola Dreyer[7,9], Christian Utpatel[7,9], Tanja Niemann[9], Andrew DiNardo[1,10], Alexander Kay[1,5], Stefan Niemann[7,9], Anna Mandalakas[1,5,6,7,10]

**1** Global Tuberculosis Program, Baylor College of Medicine, Houston, Texas, United States of America, **2** Department of Biological Sciences, University of Alaska, Anchorage, Alaska, United States of America, **3** Eswatini Health Laboratory Service, Ministry of Health, Mbabane, Eswatini, **4** Department of Biomedical Sciences, Cape Peninsula University of Technology, Bellville, South Africa, **5** Baylor College of Medicine Children's Foundation-Eswatini, Mbabane, Eswatini, **6** Department of Clinical Infectious Diseases, Research Center Borstel, Leibniz Lung Center, Borstel, Germany, **7** German Center for Infection Research, Partner Site Hamburg-Lübeck-Borstel-Riems, Borstel, Germany, **8** Respiratory Medicine & International Health, University of Lübeck, Lübeck, Germany, **9** Research Center Borstel – Leibniz Lung Center, Molecular and Experimental Mycobacteriology, Borstel, Germany, **10** William T Shearer Center for Immunobiology, Texas Children's Hospital, Baylor College of Medicine, Houston, Texas, United States of America

☯ These authors contributed equally to this work.
* tara.ness@bcm.edu

## Abstract

In 2023, an estimated 10.8 million people developed tuberculosis, and 1.25 million people died from this disease, including 161,000 deaths in people with HIV (PWH) in whom tuberculosis remains the leading cause of death. Detecting *Mycobacterium tuberculosis* drug resistance remains a challenge among patients with paucibacillary tuberculosis; since there is such low bacterial load in their sputum it's unable to be detected via microscopy, there is also not enough bacteria for other sputum-based tests which could provide resistance testing. At an outpatient clinic in Eswatini from 2020-2023, stool and sputum samples were provided by a subset of children, adolescents, and adults prospectively enrolled in a tuberculosis diagnostic study. In addition to standard diagnostic testing available in country (direct sputum Xpert, stool Xpert, and phenotypic drug susceptibility testing of sputum culture), stool samples underwent extraction and sequencing using targeted next generation sequencing (tNGS), using both the Oxford Nanopore Technologies (ONT) TB Custom Kit (on an ONT MinION Mk1b) and the Deeplex Myc-TB kit (on an Illumina iSeq 100). From 250 participants with pulmonary tuberculosis diagnosed in Eswatini during our study period, 85 (34%) were smear negative on sputum microscopy. Of these, 21/85 (24.7%) participants had adequate *M. tuberculosis* DNA shed in their stool for attempting

which permits unrestricted use, distribution, and reproduction in any medium, provided the original author and source are credited.

**Data availability statement:** Our data is available at the National Center for Biotechnology Information (of the National Institutes of Health) Sequence Read Archive under PRJNA1273636.

**Funding:** This work has been funded by the US National Institutes of Health (R01AI137527 to AM and 1K01TW011482-01A1 to AK), the Thrasher Foundation (14323 to AK and 01098 to TN), the St. Jude's/PIDS Award in Basic and Translational Science (to T Ness), and Sub-Saharan SeqNet (Global Health Protection Program (GHPP) supported by the Federal Ministry of Health, German Bundestag 2523GHP013 to CL). CL is supported by the German Center for Infection Research (DZIF) under grant agreement TTU-TB 02.709. The funders had no role in study design, data collection and analysis, decision to publish, or preparation of the manuscript. Of note, the Oxford Nanopore Technologies OND-CUST-KIT was provided free of cost under a material transfer agreement with Baylor Center of Excellence, Mbabane, Eswatini as the kits were authorized only for research use and were not commercially available. All other sequencing reagents used were purchased at cost from the manufacturers.

**Competing interests:** The authors have declared that no competing interests exist.

tNGS. Targeted sequencing on stool detected *M. tuberculosis* DNA in 14–19% (n = 12/85–16/85) and provided a full report of mutations associated with drug resistance in 12–14% (n = 10/85–12/85) of patients with paucibacillary (smear-negative) tuberculosis, expanding drug resistance detection beyond other methods. Targeted sequencing of stool, even when applied to patients with paucibacillary disease, can provide case confirmation and expanded drug resistance information.

## Background

In 2023, an estimated 10.8 million people developed tuberculosis and 1.25 million people died from this disease, including 161,000 deaths in people with HIV (PWH) in whom tuberculosis remains the leading cause of death [1]. Drug-resistant tuberculosis (DR-tuberculosis) made up 3.2% of newly diagnosed cases and 16% of previously treated cases [1]. Eswatini continues to be one of the highest burdened countries by tuberculosis, with a tuberculosis incidence of 350 per 100,000 and multidrug or rifampin resistant TB (MDR/RR-TB) incidence of 18 per 100,000 [1]. Targeted next generation sequencing (tNGS) has shown to improve the diagnosis of tuberculosis [2] when applied on samples collected at the point-of-care (including direct sputum [3] and stool [4]). Targeted NGS can extend the identification of mutations conferring resistance to antibiotics used to treat tuberculosis, including antibiotics for which phenotypic drug susceptibility testing (pDST) is not commonly available in high-burden settings [3].

Thirty to 60% of individuals with pulmonary tuberculosis disease are smear-negative (have undetectable acid-fast bacilli (AFB) on sputum microscopy) [5], defined as paucibacillary disease, which often applies especially to children and PWH. Detection of AFB on sputum microscopy is used in many high-burden countries as an initial diagnostic test, with the World Health Organization (WHO) reporting more than half of tuberculosis diagnoses being made clinically or via sputum microscopy [1]. When tNGS is performed directly on sputum it can increase case confirmation by more than 10% compared to sputum smear microscopy alone [6] and yield at least partial sequencing results in 62–78% of sputum culture positive adults with paucibacillary disease [7]. Recently, tNGS has been successfully applied to stool [4] highlighting the potential for important additional case finding in individuals with paucibacillary disease, especially in young children who are often unable to produce sputum. The WHO has previously endorsed the use of stool for tuberculosis diagnosis in children using the Xpert MTB/Rif test [3] and, in the year 2023, the WHO endorsed tNGS [8] as a cost-effective [9] and adaptable diagnostic tool [10] to predict *Mycobacterium tuberculosis* drug resistance

To ascertain the added benefit of tNGS performed on stool for the diagnosis of paucibacillary pulmonary tuberculosis and for comprehensive prediction of *M. tuberculosis* drug susceptibility, we performed a prospective diagnostic accuracy study among participants with undetectable AFB on sputum microscopy ("smear negative") in Eswatini, a tuberculosis high-burden country.

## Methods

### Ethics statement

Clinical investigations were conducted according to the principles expressed in the Declaration of Helsinki with written informed consent obtained from all participants. Approval was obtained from all necessary ethical bodies including the Baylor College of Medicine Children's Foundation Eswatini (IORG0006978) and the Eswatini National Human Health Research Review Board, Baylor College of Medicine Institutional Review Board (FWA-00000286), Houston, Texas, USA, and the Cape Peninsula University of Technology Health and Wellness Sciences Research Ethics Committee (CPUT/HWS-REC 2024/H10), Cape Town, South Africa.

### Study population

Stool samples were provided by a subset of children, adolescents, and adults prospectively enrolled in a tuberculosis diagnostic study taking place from October 1, 2020 to December 31, 2024 as previously described [11]. Existing patient data was accessed for this analysis in January 2025. Participants were recruited from the outpatient tuberculosis clinic of the Baylor College of Medicine Children's Foundation Eswatini in Mbabane, Eswatini, and continue to participate in longitudinal follow-up for capture of treatment and clinical outcomes. All participants in this study were diagnosed with pulmonary tuberculosis per standard in-country testing and clinical standards.

### DNA extraction and quantitative polymerase chain reaction

Stool samples were stored at -80°C within 12 hours of collection and DNA extraction was done in batches from frozen aliquots using the MP Fast DNA kit for soil (MP Biochemicals, Solon, OH) as previously described [12]. Quantitative polymerase chain reaction (qPCR) was completed on stool DNA extracts to determine the relative amount of *M. tuberculosis* DNA using a previously validated protocol incorporating primers targeting the repetitive IS6110 region of the *M. tuberculosis* genome [11,13]. Participants were defined as shedding *M. tuberculosis* in their stools if *M. tuberculosis* DNA was detected at a Cycle Threshold (Ct) less than 33 (approximately 1 fg/µl of *M. tuberculosis* DNA) [13] based on previous studies [4]. All laboratory testing procedures were done on-site in Eswatini.

### Sequencing

Stool DNA extracts underwent tNGS using both 1) the Deeplex Myc-TB kit (Genoscreen, Lille, France) on an Illumina iSeq 100 (Illumina Inc, San Diego, CA, USA), and 2) the OND-CUST-KIT (Oxford Nanopore Technologies [ONT], Oxford, UK) on an ONT MinION sequencer. Stool DNA was prepared for sequencing based on the manufacturer's instructions. For the Deeplex Myc-TB kit, stool DNA was prepared using the Baym library preparation protocol [14], a modified Nextera XT kit (Illumina Inc, San Diego, CA, USA) and sequenced with 150 base pairs (bp) paired-end reads using iSeq 100 i1 Reagent v2 (300-cycle) on an iSeq100 instrument. Analyses were performed using the integrated bioinformatics pipeline, Deeplex Myc-TB V3_0_1-extended catalogue on the Deeplex® Myc-TB web application (https://euc1.platform.illumina.com/Deeplex). Reads were mapped and called at 3% detection threshold for drug resistance detection. For the OND-CUST-KIT, stool DNA underwent targeted amplification with the Invitrogen Platinum II Taq Hot-Start DNA Polymerase (Fisher Scientific, Hampton, NH, USA) with library preparation and barcoding performed using the ONT-Rapid Barcoding Kit 96 (SQK-RBK110–96) and run on an R9 flow cell (FLO-MIN106D) per manufacturer's instructions. Nanopore sequencing, base-calling, and de-multiplexing including the generation of FASTQ files were done with ONT MinKnow software on an ONT MinION Mk1b sequencer. Further analysis was performed with EPI2ME Labs using the ONT automated pipeline 'wf-tb-amr' (workflow [v2.0.0-alpha.4], minimap2 [2.24-r1122], samtools [1.18], bedtools [v2.30.0], bcftools [1.18]) with any discrepancies between sequencing and other diagnostic testing additionally run through TB Profiler [15].

## Other diagnostic testing

Sputum was evaluated by Xpert Ultra (Cepheid, Sunnyvale, CA, USA) and liquid culture (BACTEC Mycobacteria Growth Indicator Tube [MGIT] system; Becton Dickinson, Franklin Lakes, NJ, USA). Sputum was cultured at the National Tuberculosis Reference Laboratory in Mbabane, Eswatini using MGIT 960 including pDST for rifampin (RIF), isoniazid (INH), and ethambutol (EMB) per national guidelines and in alignment with WHO critical concentrations. When indicated (RIF resistant cases), cultures are also tested for susceptibility to levofloxacin (LFX) and ofloxacin (OFX). Xpert Ultra was used on sputum specimens in accordance with the manufacturer's instructions; invalid results were repeated, and in case of a second error, were excluded. Stool samples were tested by Xpert Ultra after being processed using the simple one-step stool processing method [16].

## Data analysis

Individuals were included in the analysis if they were diagnosed with pulmonary tuberculosis disease and had a negative sputum smear (undetectable AFB smear/paucibacillary tuberculosis).

Confirmed pulmonary tuberculosis disease was defined by signs/symptoms of tuberculosis along with any of the following microbiologic confirmation consistent with in-country testing and clinical standards:

a)  detection of *M. tuberculosis* in sputum via Xpert Ultra or sputum culture,

b)  detection of lipoarabinomannan [LAM] in urine in PWH [17],

c)  detection of *M. tuberculosis* in stool via Xpert Ultra in children under 15 years of age [18] or quantitative polymerase chain reaction (research use only).

Data analysis was performed in Microsoft Excel and R (version 4.4.0) to generate percentages, proportions, and statistical descriptions of demographic data (median, interquartile ranges). As this was a retrospective data analysis, participants already had a diagnosis of pulmonary tuberculosis, so statistical analysis was presented as case confirmation. The study protocol was approved by Baylor College of Medicine Children's Foundation Eswatini (IORG0006978), Eswatini National Human Health Research Review Board, and Baylor College of Medicine Institutional Review Board (FWA-00000286). All study participants and/or their legal guardians provided written informed consent. Participants 10–17 years provided written assent. Stool tNGS was not used for clinical management decisions.

## Results

From 250 participants with pulmonary tuberculosis diagnosed in Eswatini during our study period, 85 (34%) had a negative sputum smear. Of these, 21/85 (24.7%) participants had adequate *M. tuberculosis* DNA shed in their stool for tNGS per previous studies and underwent sequencing [4] Participant ages were a median of 32 years (rang 0–68 years) with 52.4% female (n = 11) and 57.1% (n = 12) living with HIV. Chest radiography was abnormal (n = 19), normal (n = 1), and not completed (n = 1) and sputum culture was negative in 38% (n = 8/21). Of those with a negative sputum culture (n = 8), 7 had a positive Xpert Ultra on sputum at baseline and one was clinically diagnosed (4 months of age at time of diagnosis; abnormal chest radiography, positive exposure, positive symptom screen) and was shedding *M. tuberculosis* DNA in their stool identified via qPCR.

Nanopore sequencing (ONT) provided detection of *M. tuberculosis* in stool from 16/21 (76%) participants with a complete drug resistance report in 12/21 (57%), providing an 18.8% (n = 16/85) increased case confirmation and a full determination of mutations associated with drug resistance in 14.1% (n = 12/85) of patients with paucibacillary tuberculosis (Fig 1). Deeplex sequencing provided detection of *M. tuberculosis* in stool from 12/20 (60%) participants with a full drug resistance report generated in 10/20 (50%), providing a 14.1% (n = 12/85) increased case confirmation for paucibacillary ("smear-negative") tuberculosis and full determination of mutations associated with drug resistance in 11.8% (n = 10/85).

# pDST sputum culture

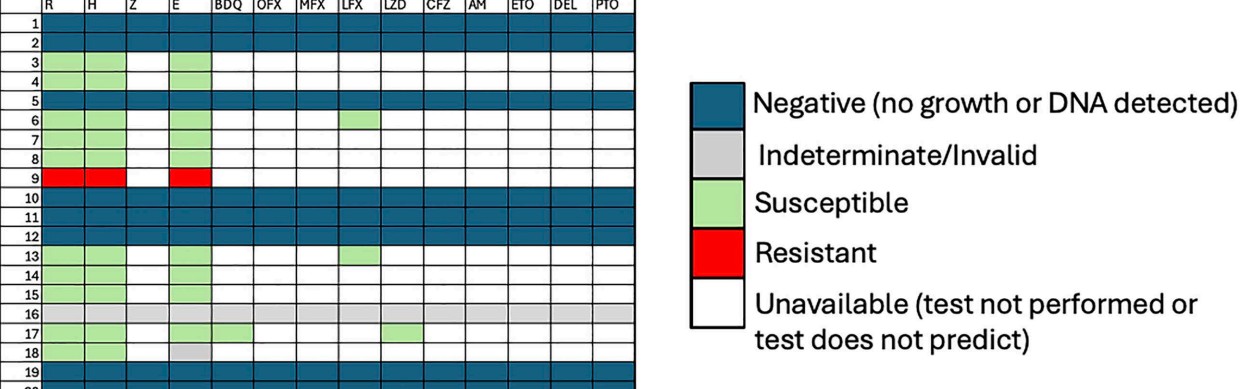

# Xpert Ultra sputum

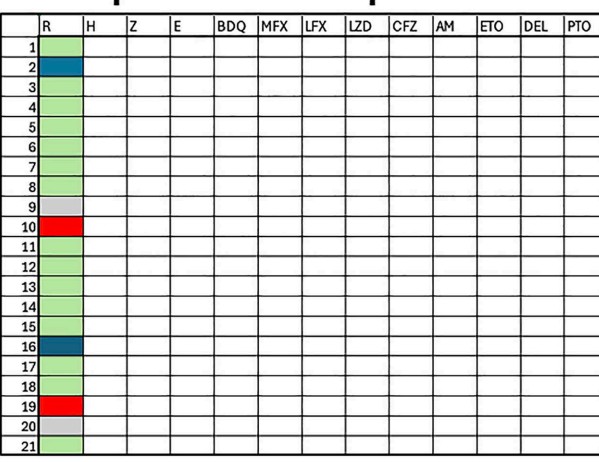

# Xpert Ultra stool

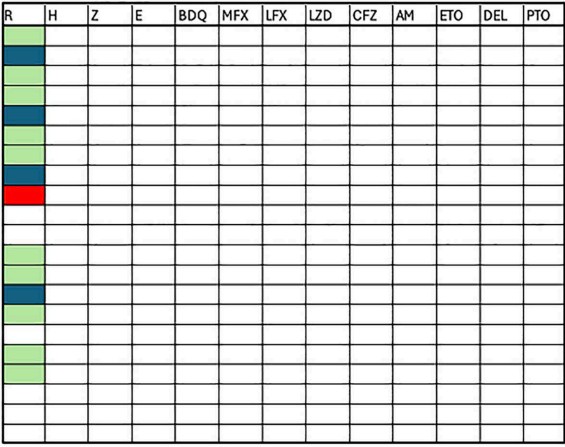

# Nanopore tNGS stool

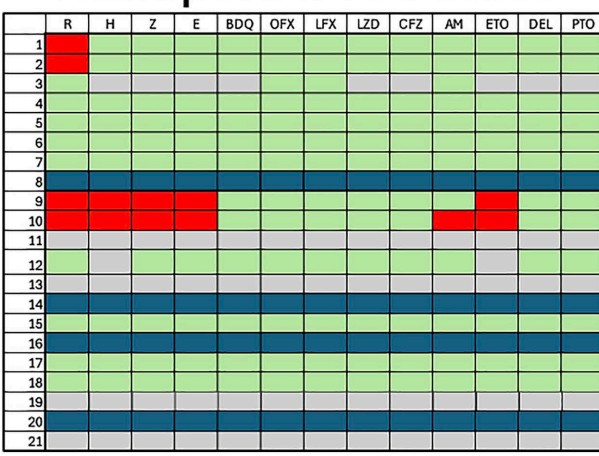

# Deeplex tNGS stool

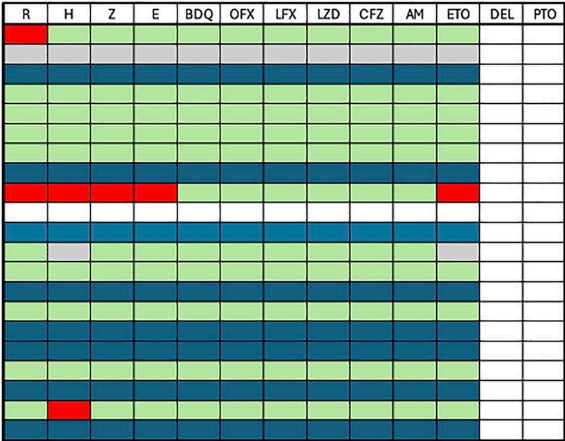

**Fig 1. Comparison of diagnostic testing at baseline for individuals diagnosed with pulmonary tuberculosis who were sputum smear negative.** *Indeterminate/Invalid for tNGS indicates that M. tuberculosis DNA was detected by targeted sequencing, however drug resistance predictions were*

*unable to be made due to the low quality/quantity of DNA in the sample. Case 9 rifampin resistance was detected using the ONT custom kit amplicons and TB Profiler but was not identified by the ONT wf-tb-amr workflow. Case 10 did not have enough sample for both nanopore and Deeplex tNGS, hence why Deeplex tNGS on stool was unable to be performed. R = rifampin; H = isoniazid; Z = pyrazinamide; E = ethambutol; BDQ = bedaquiline; OFX = ofloxacin; MFX = moxifloxacin; LFX = levofloxacin; LZD = linezolid; CFZ = clofazimine; AM = amikacin; ETO = ethionamide; DEL = delamanid; PTO; pretomanid.*

One case did not have enough sample for both ONT and Deeplex sequencing (Case 10 in Fig 1) so did not have Deeplex testing performed. Sputum culture pDST identified only one individual with drug resistance; however, drug resistance was identified via Xpert Ultra on sputum in two cases, via Xpert Ultra on stool in one case, and via nanopore and Deeplex tNGS on stool in four cases and three cases, respectively.

Rifampin resistance was identified by sputum culture in one case, Xpert Ultra sputum in two cases, and Xpert Ultra stool in one case (Fig 1). Targeted NGS by ONT identified four cases of rifampin resistance (two cases of His445Asp and two cases of Ser450Leu on the rpoB gene) and Deeplex identified two cases of rifampin resistance (one case of His445Asp and one case of Ser450Leu), identifying resistance found on other diagnostic approaches as well as providing additional identification. One case (participant 9, Fig 1) originally showed rifampin resistance on stool Xpert Ultra but not on nanopore tNGS, however this was identified on TB Profiler with 18x depth of coverage with 100% of reads showing a Ser450Leu mutation in the rpoB region. There were no other discordant determinations of drug resistance or mutations associated with drug resistance between testing modalities; however many cases of drug resistance were identified by one modality with another modality having "no growth" or "no tuberculosis DNA detected."

Three individuals in our study were under five years of age (participants 1, 2 and 8, Fig 1). Of these children, sputum culture had no growth in 2 cases, with nanopore tNGS on stool providing detection of *M. tuberculosis* DNA and full drug resistance prediction in both and Deeplex providing detection of *M. tuberculosis* DNA in both and full drug resistance prediction in one.

Four cases (participants 9, 10, 11 and 19, Fig 1) were initiated on a DR-TB regimen at baseline and two cases were lost to follow up (participants 7 and 9, Fig 1). Targeted NGS on stool provided complete or near complete drug resistance predictions in 62.5% (n = 5/8) and 40% (n = 4/8) of those with no sputum culture growth using ONT and Deeplex, respectively.

## Discussion

This study assessed the ability of stool-based targeted sequencing by two commercially available assays for the diagnosis of paucibacillary pulmonary tuberculosis and the prediction of *M. tuberculosis* drug resistance in a tuberculosis high burden country. Targeted NGS using Deeplex and ONT kits provided increased case confirmation of 14.1% and 18.8%, respectively, and full drug resistance predictions in 11.8% and 14.1%, respectively, of patients with smear-negative tuberculosis. This included participants with HIV, those with negative sputum culture growth, and several participants under five years of age; populations who could especially benefit from this technology and sample type. This study documents that stool-based tNGS could provide timely confirmation (<8 hours from sample collection to drug resistance report) of tuberculosis and initiation of effective treatment regimens in persons affected by paucibacillary tuberculosis, as stool can be obtained at initial diagnosis and has the capability to provide drug resistance predictions.

Previous studies have validated the utility of tNGS performed on stool samples pre-screened (shedding *M. tuberculosis* in their stools > 1 fg/μl of *M. tuberculosis* DNA) to optimize the performance and ensure resource conservation [4]. Our study focused on individuals with paucibacillary pulmonary tuberculosis who were shedding *M. tuberculosis* DNA in their stool (25% in our cohort). While previous studies on direct sputum have shown higher detection and prediction of drug resistance in participants with undetectable AFBs on sputum microscopy (62–78% compared to our study with 14 to 18.8%), it is important to note the inclusion criteria required the previous study participants to have a positive GeneXpert

result on direct sputum AND have *M. tuberculosis* detectable on sputum cultures. Further, their study excluded children and included few (14%) participants with HIV, [7] which are important populations who stand to benefit from tNGS to predict *M. tuberculosis* drug resistance from stool.

This study is limited by not pairing tNGS results from stool with results of whole genome sequencing (WGS) of sputum culture or tNGS on direct sputum. Nevertheless, nearly 40% of individuals with undetectable AFB on sputum who qualified for tNGS were persons with undetectable *M. tuberculosis* culture from sputum, thereby making WGS impossible (and previous studies have already validated the agreement between tNGS on stool and WGS of sputum culture isolates).[4] As there is emerging evidence demonstrating the potential of tNGS on direct sputum for paucibacillary disease [7], this study would have benefitted from comparison to tNGS on direct sputum; which will be incorporated into future studies. It's important to note there was one discrepancy noted betwee[n testing modalities, which was a rifampin resistance noted on stool Xpert Ultra that was not identified by the current version of the "wf-tb-amr" workflow provided by ONT, but was identified using their amplicons using TB Profiler [15] so was a limitation of their bioinformatics but not amplicon design. The differences in detection between the two commercial kits could be due to differences in sequencing technology (nanopores detecting electrical currents compared to sequencing by synthesis) having different sensitivity rates for *M. tuberculosis* DNA in stool samples. It is also worth noting that one stool sample was of minimal volume and unable to be run on Deeplex; therefore could be contributing to the detection difference. Further research studies are needed comparing these two commercial kits on extracted stool DNA samples to delineate further.

In conclusion, application of both commercially available tNGS assays (ONT TB Custom Kit and Deeplex Myc-TB) to stool of participants with paucibacillary tuberculosis, including children and PWH, increases tuberculosis case confirmation and expanded *M. tuberculosis* drug susceptibility/resistance prediction results.

## Author contributions

**Conceptualization:** Tara Ness, Mangaliso Ziyane, Wethusonke Khumalo, Christoph Lange, Viola Dreyer, Christian Utpatel, Tanja Niemann, Andrew DiNardo, Alexander Kay, Stefan Niemann, Anna Mandalakas.

**Data curation:** Tara Ness, Mangaliso Ziyane, Nontobeko Maphalala, Abigail Seeger, Wethusonke Khumalo, Mdigo Thunzini, Bryce Inman, Viola Dreyer, Christian Utpatel, Tanja Niemann, Andrew DiNardo, Alexander Kay, Stefan Niemann.

**Formal analysis:** Tara Ness, Mangaliso Ziyane, Abigail Seeger, Christoph Lange, Bryce Inman, Viola Dreyer, Christian Utpatel, Andrew DiNardo, Stefan Niemann.

**Funding acquisition:** Tara Ness, Alexander Kay.

**Investigation:** Tara Ness, Mangaliso Ziyane, Nontobeko Maphalala, Anca Vasiliu, Christoph Lange, Tanja Niemann, Andrew DiNardo, Alexander Kay, Anna Mandalakas.

**Methodology:** Tara Ness, Mangaliso Ziyane, Nontobeko Maphalala, Anca Vasiliu, Clement Gascua, Christoph Lange, Viola Dreyer, Tanja Niemann, Andrew DiNardo, Alexander Kay, Anna Mandalakas.

**Project administration:** Andrew DiNardo, Anna Mandalakas.

**Resources:** Andrew DiNardo.

**Supervision:** Sindisiwe Dlamini, Gugu Maphalala, Clement Gascua, Christoph Lange, Samantha Meyer, Andrew DiNardo, Alexander Kay, Stefan Niemann.

**Validation:** Tara Ness, Andrew DiNardo, Anna Mandalakas.

**Visualization:** Andrew DiNardo.

**Writing – original draft:** Tara Ness, Mangaliso Ziyane, Anna Mandalakas.

**Writing – review & editing:** Tara Ness, Mangaliso Ziyane, Nontobeko Maphalala, Abigail Seeger, Anca Vasiliu, Wethusonke Khumalo, Clement Gascua, Christoph Lange, Bryce Inman, Viola Dreyer, Christian Utpatel, Tanja Niemann, Andrew DiNardo, Alexander Kay, Stefan Niemann, Anna Mandalakas.

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
