## [Decision Letter · Decision Letter 0]

13 May 2025

PGPH-D-25-00536

Screening and Targeted Sequencing of Stool for Microbiologic Confirmation and Drug Resistance Determination in Paucibacillary Tuberculosis (Brief Report)

Dear Dr. Ness,

Thank you for submitting your manuscript to PLOS Global Public Health. After careful consideration, we feel that it has merit but does not fully meet PLOS Global Public Health’s publication criteria as it currently stands. Therefore, we invite you to submit a revised version of the manuscript that addresses the points raised during the review process.

We look forward to receiving your revised manuscript.

Kind regards,

Cesar Ugarte-Gil, MD, MSc, PhD

Academic Editor

Journal Requirements:

Additional Editor Comments (if provided):

Reviewers' comments:

Reviewer's Responses to Questions

**Comments to the Author**

1. Does this manuscript meet PLOS Global Public Health’s publication criteria?

Reviewer #1: Yes

Reviewer #2: Yes

Reviewer #3: Partly

2. Has the statistical analysis been performed appropriately and rigorously?

Reviewer #1: Yes

Reviewer #2: Yes

Reviewer #3: No

3. Have the authors made all data underlying the findings in their manuscript fully available (please refer to the Data Availability Statement at the start of the manuscript PDF file)?

Reviewer #1: Yes

Reviewer #2: Yes

Reviewer #3: Yes

4. Is the manuscript presented in an intelligible fashion and written in standard English?

Reviewer #1: No

Reviewer #2: Yes

Reviewer #3: No

Reviewer #1: I have reviewed the manuscript by Ness et al entitled "Screening and Targeted Sequencing of Stool for Microbiologic Confirmation and Drug Resistance Determination in Paucibacillary Tuberculosis (Brief Report)". This is an informative manuscript in an area less researched. I have highlighted areas for improvement below:

1) The abstract is too brief and seems not to be representing the entire manuscript.

2) Background: The burden of TB and diagnostic challenges as per the study setting and country are missing. The 2nd sentence under background is too long, please consider revising the writing style throughout the manuscript.

3) Thirty to 60% of individuals affected by pulmonary tuberculosis... Is its affected or infected, better to use the term "presumptive TB patients"

4) Under last part of background: "we performed a prospective diagnostic accuracy study with participants with undetectable AFB " please revise ..accuracy study "AMONG" participants . Also, the term "undetectable AFB" is confusing, please use smear negative or negative for the primary test.

5) Under methods section "Data was accessed for analysis January 2025., very brief sentence and not clear, please revise

6) ....Foundation Eswatini in Mbabane, Eswatini, and continue to participate in longitudinal follow-up, This is not clear, please revise

7) ..Pulmonary tuberculosis was defined by a) detection of M. tuberculosis in sputum, and/or b) detection of M. tuberculosis DNA by Xpert Ultra on sputum or stool, and/or c) detection of lipoarabinomannan [LAM] in urine in PWH. Please consider making this section clearer 1) define all terms used in standard way, including undetectable AFB, if you choose to maintain it, 2) LAM among PLHIV, this is a conditional test and it should not be used uniformly but rather guided by CD4 and whether the patient was admitted, 3) Is stool GeneXpert test a standard test for TB diagnosis among adult PLHIV presumptive TB patients in Eswatini? if not, it can not be used in the definition of a case in such population, 4) Please elaborate more on the data analysis and possible adding a sub section to that, 5) please define the "paucibacillary' nature of this study population

8) Please consider more sub sections as required by the journal under the methods section.

9) Under Results section: Participant ages ranged from 0 to 68 years (median 32 years).. Please consider presenting median age ( IQR) in a standard way

10) Throughout the results section, please add percentages to all proportions.

11) "Targeted NGS on stool provided complete or near complete drug resistance predictions"...What do you mean by "prediction "consider using "detection "or define what you mean by prediction under methods section.

12) Under discussion" Targeted NGS using Deeplex and nanopore provided increased case confirmation of 14.1% and 18.8%, respectively, and full drug resistance predictions in 11.8% and 14%, respectively. Please discuss the possible explanations of this difference in detection rates between Nanopore and Deeplex.

13) "Further, their study excluded children and included few (14%) participants with HIV,7 which are important populations who stand to benefit from tNGS to predict M. tuberculosis drug susceptibility and resistance from stool" What is the difference between drug susceptibility and resistance? please revise.

14) In conclusion, application of both commercially available tNGS assays to stool of participants with paucibacillary tuberculosis" Please mention the tNGS assays since they are just two i.e. Nanopore and Deeplex.

Reviewer #2: Summary

Ness and colleagues have investigated the use of 2 commercially available tNGS assays, Deeplex and Nanopore, on stool samples from 21 participants with TB. Study participants included children and adults and all were AFB smear negative. This brief report provides some evidence for the utility of tNGS on stool samples, which is a promising finding as people with TB who cannot undergo traditional DST methods can still benefit from receiving Mtb confirmation and drug resistance testing. Some specific comments are provided below, particularly to improve clarity and consistency of reporting.

Specific comments

Abstract: I imagine the word count for a brief report abstract is very tight, but as it stands the abstract does not provide enough information to know what happened in the project. Could the abstract be revised to include the study objective, test/samples, and participant population? Something similar to the stated objective at the end of the introduction would work well. Additionally, the meaning of the phrase “…can detect Mtb DNA in 14-19%” is unclear. 14-19% of what? All patients with TB? All patients with paucibacillary TB? all tested patients? Please clarify in-text.

Methods: Description of lab testing procedures is quite clear. Could you please clarify if all lab procedures and index test were run on-site in Mbabane?

Methods: Can you please provide further details regarding how participants were recruited for this particular study? Were all participants known to have TB before being recruited into the study? Were there any participants who did not have TB who were included in the study? Or was it a presumptive series of people seeking care? Something else?

Methods: As written, it is a bit difficult to understand who this project’s participants are. Please include details such as the inclusion and exclusion criteria, e.g., age, HIV status, smear status. How was paucibacillary/undetectable TB defined? Did all 21 people meet this definition?

Results: among the 8/21 PTB+ individuals who were sputum culture negative, how was TB diagnosed?

Results: The phrase “…providing a 14.1% (n=12/85) increased case confirmation for paucibacillary tuberculosis…” is unclear to me. What is the comparison of “increased” being made to here? Smear microscopy? Microbiological confirmation? Something else? From earlier in the Methods/Results, I thought that everyone in the study had to have a positive sputum, stool, or urine test already – is it the case that some people did not meet this criteria?

Discussion: Please consider modifying the claim in the discussion that “This study documents that stool-based tNGS improves timely confirmation of tuberculosis and initiation of effective treatment regimens…” There are no results presented here regarding time to results availability or time to treatment initiation.

Discussion: Is the group referred to in “Nevertheless, nearly 40% of individuals with undetectable AFB on sputum who qualified for tNGS were persons with undetectable M. tuberculosis culture from sputum, thereby making WGS impossible” the same 34% of individuals who were previously stated as having “undetectable AFB on sputum microscopy” in the Results? If so, I would also include in the Results section that the 34% were also culture-negative, to make it more obvious why they didn’t undergo sputum tNGS.

Conclusion: yes, agree, especially that expanded drug susceptibility testing is a benefit!

Reviewer #3: This brief report presents the results of a prospective diagnostic accuracy study conducted in Eswatini, a high tuberculosis (TB) burden country, involving participants with negative acid-fast bacilli (AFB) results on sputum microscopy.

Although the World Health Organization (WHO) recommended the use of stool specimens for diagnosing Mycobacterium tuberculosis complex (MTBC) in children using the Cepheid Xpert MTB/RIF assay in 2020—a time frame that coincides with this study—the authors did not reference this guidance in the background, nor in the discussion sections of the report.

The study includes an image displaying results from phenotypic drug susceptibility testing (pDST), Xpert testing on sputum and stool, and targeted next-generation sequencing (tNGS) on stool using both Nanopore and Deeplex platforms. However, the authors have only analyzed the added diagnostic value of tNGS compared to microscopy. They have not compared the diagnostic yield of stool Xpert with that of tNGS. Furthermore, rifampicin results and resistance discordances between different specimen types and diagnostic methods have not been assessed. (See attached file)

The Authors should address all these issues in revised version

**Do you want your identity to be public for this peer review?** For information about this choice, including consent withdrawal, please see our Privacy Policy

Reviewer #1: **Yes: ** Willy Ssengooba

Reviewer #2: No

Reviewer #3: No

---

## [Decision Letter · Decision Letter 1]

20 Oct 2025

PGPH-D-25-00536R1

Screening and Targeted Sequencing of Stool for Microbiologic Confirmation and Drug Resistance Determination in Paucibacillary Tuberculosis

Dear Dr. Ness

Thank you for submitting your manuscript to PLOS Global Public Health. After careful consideration, we feel that it has merit but does not fully meet PLOS Global Public Health’s publication criteria as it currently stands. Therefore, we invite you to submit a revised version of the manuscript that addresses the points raised during the review process.

We look forward to receiving your revised manuscript.

Kind regards,

Wilber Sabiiti

Academic Editor

Journal Requirements:

Additional Editor Comments (if provided):

I agree with Reviewer 3 that a diagram summarising RIF results across the tests (if data available) would be enhance the manuscript. In the abstract, I understand you are trying to keep to the word limit but the ranges on tGS results are quite confusing. "Targeted sequencing on stool detected M. tuberculosis DNA in 14-19% (n=12/85 to 16/85) and provided a full report of mutations associated with drug resistance in 12-14% (n=10/85 to 12/85) of patients with paucibacillary (smear-negative) tuberculosis, expanding drug resistance detection beyond other methods". Instead of ranges, would it be possible to mention the results per tGS method so that readers know right away the method responsible for detecting less or more drug resistance.

Reviewers' comments:

Reviewer's Responses to Questions

**Comments to the Author**

Reviewer #1: All comments have been addressed

Reviewer #2: All comments have been addressed

Reviewer #3: All comments have been addressed

publication criteria?

Reviewer #1: Yes

Reviewer #2: Yes

Reviewer #3: Yes

3. Has the statistical analysis been performed appropriately and rigorously?

Reviewer #1: Yes

Reviewer #2: Yes

Reviewer #3: N/A

4. Have the authors made all data underlying the findings in their manuscript fully available (please refer to the Data Availability Statement at the start of the manuscript PDF file)?

Reviewer #1: Yes

Reviewer #2: Yes

Reviewer #3: Yes

5. Is the manuscript presented in an intelligible fashion and written in standard English?

Reviewer #1: Yes

Reviewer #2: Yes

Reviewer #3: Yes

Reviewer #1: All my comments have been addressed.

Reviewer #2: Thanks to the authors for addressing my comments. I think the flow of information has been strengthened, and the improvement in study population details and findings will certainly be of interest to readers. It is a lot easier to follow now and understand the significance of results. nice work! I have no further substantive comments to make.

Please have a careful read through the paper for grammar and clarity. Please remove contractions and check consistency of terminology.

Reviewer #3: The revised manuscript is well-structured, and the authors have satisfactorily addressed all the reviewers’ comments. However, the manuscript would benefit from the inclusion of a clear figure or Table comparing RIF results across the different diagnostic methods used. This addition would enhance clarity and support the interpretation of findings.

**Do you want your identity to be public for this peer review?** For information about this choice, including consent withdrawal, please see our Privacy Policy

Reviewer #1: **Yes: ** Willy Ssengooba

Reviewer #2: No

Reviewer #3: No

---

## [Editor Report · Decision Letter 2]

3 Nov 2025

Screening and Targeted Sequencing of Stool for Microbiologic Confirmation and Drug Resistance Determination in Paucibacillary Tuberculosis

PGPH-D-25-00536R2

Dear Dr Ness

We are pleased to inform you that your manuscript 'Screening and Targeted Sequencing of Stool for Microbiologic Confirmation and Drug Resistance Determination in Paucibacillary Tuberculosis' has been provisionally accepted for publication in PLOS Global Public Health.

Best regards,

Wilber Sabiiti

Academic Editor

Authors have addressed all the queries. Please follow the journal guidelines to finalise your manuscript for publication. Some of figures have poor resolution, please improve them for publication.